# σReparam: Stable Transformer Training with Spectral Reparametrization

## Abstract

Training stability is of great importance to Transformers. In this work, we investigate the training dynamics of Transformers by examining the evolution of the attention layers. In particular, we track the "attention entropy" for each attention head during the course of training, which is a proxy of the attention's sharpness. We observe a common, non monotonic evolution of attention entropy across different settings: the attention entropy first quickly decreases in the initial phase of training, followed by quickly increasing, and finally entering a long stable phase. While the exact shape can be affected by hyperparameters such as warmup, initialization, learning rate etc., we found that there is a close correlation between the minima of attention entropy and the model's training stability. To this end, we propose a simple and efficient solution dubbed σReparam, where we reparametrize all linear layers with Spectral Normalization and an additional learned scalar. We provide a lower bound on the attention entropy as a function of the spectral norms of the query and key projections, which suggests that small attention entropy can be obtained with large spectral norms. σReparam decouples the growth rate of a weight matrix's spectral norm from its dimensionality, which we verify empirically. We conduct experiments with σReparam on image classification, image self supervised learning, automatic speech recognition and language modeling tasks. We show that σReparam provides great stability and robustness with respect to the choice of hyperparameters.

## 1 Introduction

Transformers (Vaswani et al., 2017) are state-of-the-art models in many application domains. However, despite their empirical success and wide adoption, great care often needs to be taken in order to achieve good training stability and convergence. In the original paper (Vaswani et al., 2017), residual connections and Layer Normalizations (LNs) (Ba et al., 2016) are extensively used for each Attention and MLP block (specifically, in the "Post Norm" fashion). There has since been various works attempting to promote better training stability and robustness. For example, the "Pre Norm" (Radford et al., 2019) scheme has gained wide popularity, where one moves the placement of LNs to the beginning of each residual block. Others have argued that it is important to properly condition the residual connections. Bachlechner et al. (2021) proposes to initialize the residual connections to zero to promoter better signal propagation. Zhang et al. (2018); Huang et al. (2020) remove LNs with carefully designed initialization schemes.

In this work, we study the training instability of Transformers from the lens of training dynamics. We start by monitoring the average entropy of the attention heads (by treating each attention head as a multinomial distribution) over all query positions and examples. Interestingly, the average attention entropy often evolves in a pattern consisting of three phases. In the beginning, attention entropy starts high (corresponding to uniform attention scores) and quickly drops to a small value; This is then followed by a second stage where it quickly increases to a relatively high entropy regime; Lastly the attention entropy curve stabilizes and smoothly evolves to convergence. See the top left plot of Figure 1 for an illustration, which is a Vision Transformer (Touvron et al., 2021) (ViT) trained on ImageNet classification, using well optimized hyper parameters.

Empirically, we have found that the attention entropy is directly correlated with the model's stability and convergence. In particular, small attention entropy reached in the initial phase often causes slow

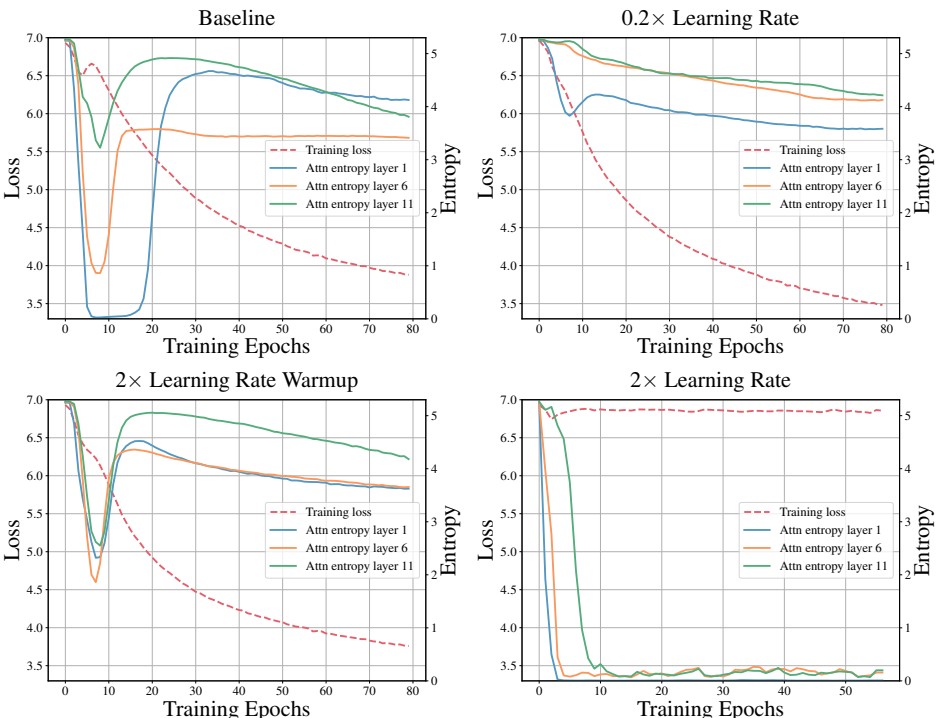

Figure 1: The training loss curves of ViT-B on ImageNet, together with the attention entropy for three layers. From top left to bottom right: baseline with default hyper parameters from Touvron et al. (2021); $0.2\times$ learning rate; $2\times$ warmup epochs; $2\times$ learning rate. We see a close correlation between the dip of the attention entropy and the convergence and stability of the training loss.

convergence, fluctuations in training loss and, in the worst case, divergence. This is shown in Figure 1 where we vary the learning rate and warmup epochs of the baseline ViT model. We see that both decreased the learning rate and increased warmup epochs provide smoothing effects to the attention entropy curves, which in turn yield lower training losses. On the other hand, increasing learning rate brings a detrimental impact on training where the attention entropy collapses to near zero and training diverges. We denote the rapid dip of attention entropy to a near zero value and its resulting pathological optimization dynamics as "entropy collapse".

The remaining questions are: 1) How do we get rid of entropy collapse? 2) Can we improve training stability by doing so? We answer them by showing that attention entropy is closely related to the spectral norms of the query and key projections. In particular, we show a lower bound of the attention entropy, which suggests that large spectral norms of the projections can more easily lead to entropy collapse. We then provide a simple fix, dubbed $\sigma$Reparam, which reparameterizes all weight matrices by sequentially applying Spectral Normalization (Miyato et al., 2018) and a learned multiplicative scalar. Intuitively, $\sigma$Reparam decouples the update of the spectral norms of weights from their dimensionality, which allows them to update smoothly in a controlled way. Also note that $\sigma$Reparam does not change the model space, which allows one to learn an arbitrarily expressive model.

We validate $\sigma$Reparam on 4 tasks: image classification, image self supervised learning, automatic speech recognition and language modelling. We show that $\sigma$Reparam effectively slows down the growth of each layer's spectral norms, and as a result, their attention entropy curves are greatly smoothed. This allows us to achieve great robustness with respect to the choice of hyper parameters. In certain cases, we are able to remove Layer Norms and still achieve competitive results.

## 2 RELATED WORKS

Transformers have relied heavily on LNs to achieve training stability. Besides the popular Post Norm and Pre Norm configurations, other variants have been proposed (Wang et al., 2022; Shleifer et al.,

2021). $\sigma$Reparam does not rely on LN and can even work in the absence of it, which avoids the computational over head of explicit activation normalization.

There have also been numerous attempts to design better Transformer initialization schemes, including Zhang et al. (2018); Huang et al. (2020); Yang et al. (2022); Bachlechner et al. (2021). $\sigma$Reparam is an orthogonal approach as it addresses the training dynamics of attention layers, which makes it compatible with standard initialization methods and provides robust performance.

$\sigma$Reparam is a special case of weight reparameterization, which has found wide adoption in Deep Learning. Weight Norm (WN) (Salimans & Kingma, 2016) is a well known example of such methods, but its effectiveness in Transformers is limited. In ConvNets, simple additive weight reparameterization (Ding et al., 2021) has been demonstrated useful in speeding up training convergence. To the best of our knowledge, $\sigma$Reparam is the first simple reparamterization technique that provides competitive performance with well optimized baseline models.

## 3 METHOD

### 3.1 ATTENTION ENTROPY

At the core of Transformers is dot product attention. Let $X \in \mathbb{R}^{T \times d}$ denote an input sequence to an attention layer (we assume self attention for simplicity of presentation), where $T, d$ are the number of tokens and the token dimension, respectively; and let $W_K, W_Q \in \mathbb{R}^{d \times n_a}, W_V \in \mathbb{R}^{d \times n_v}$ denote the key, query and value matrices. A simple attention layer then computes $\text{Att}(X) = AXW_V$ where $A = \psi(a), a = XW_K W_Q^\top X^\top$ and $\psi$ is the row-wise softmax function. We define the attention entropy of a row $i$ of $A$ by $\text{Ent}(A_i) = -\sum_{j=1}^{T} A_{i,j} \log(A_{i,j})$. We also overload the notation and let $\text{Ent}(A) = \frac{1}{T} \sum_{i=1}^{T} \text{Ent}(A_i)$ denote the average attention entropy of $A$. As shown in Figure 1, the attention entropy (and the entropy collapse phenomenon) is a strong indicator of training stability of Transformers. Our goal is to alleviate the entropy collapse problem and achieve a smooth evolution of the attention entropy through training.

We next investigate the properties of attention entropy. We show in the the next theorem that $\text{Ent}(A)$ is directly connected to the Spectral norm (the largest singular value) of $W_K W_Q^\top$.

**Theorem 3.1** (Attention entropy lower bound). *Assume without loss of generality $\|X\|_2 \leq 1$, and let spectral norm $\sigma = \|W_K W_Q^\top\|_2$. Then it holds that:*

$$Ent(A_i) \geq \log \left( 1 + (T-1)e^{-\sigma\sqrt{\frac{T}{T-1}}} \right) + \frac{\sigma\sqrt{T(T-1)}e^{-\sigma\sqrt{\frac{T}{T-1}}}}{1 + (T-1)e^{-\sigma\sqrt{\frac{T}{T-1}}}} \tag{1}$$

*Moreover, there exists inputs $X$ and weights $W_K, W_Q$ for which the lower bound in Eq. (1) is tight.*

Therefore, for large $\sigma, T$, the minimum attainable entropy behaves like $\Omega(T\sigma e^{-\sigma})$. We note that the bound on the entropy in Theorem 3.1 is tight in a sense that it is achievable for some inputs $X$. Moreover, the typical Frobenious (L2) regularization would not ensure a small $\sigma$ (a small Frobenious norm is much more restrictive than a small Spectral norm), hence it would not be as effective in preventing an "entropy collapse". Proofs for Theorem3.1 and the following Proposition are provided in Appendix A.

### 3.2 $\sigma$REPARAM

We then present $\sigma$Reparam, a method to re-parameterize the weights of a linear layer with:

$$\widehat{W} = \frac{\gamma}{\sigma(W)}W, \tag{2}$$

where $\sigma(W) \in \mathbb{R}$ is the spectral norm of $W$ and $\gamma \in \mathbb{R}$ is a learnable parameter, initialized to 1. In practice, $\sigma(W)$ can be computed via power iteration (Mises & Pollaczek-Geiringer, 1929) as in Spectral Normalization (SN) (Miyato et al., 2018), see Algorithm 1 for a sketch implementation. Note that $\sigma$Reparam brings little extra overhead as the power iteration mainly consists of two matrix vector products and is only performed on the parameters rather than activations. During inference, one can compute $\widehat{W}$ once and freeze it, which means that it has the same cost as a regular linear layer.

Table 1: Supervised Image Classification on ImageNet1k. The B/L refer to ViT-B/16 and ViT-L/16 variants respectively. SN corresponds to the Spectral Norm baseline without the learnable scalar. Also note that the WN configuration leads to immediate divergence without using Layer Norm, and here we only report the result with WN + LN.

|  | DeiT (B) | $\sigma$Reparam (B) | SN (B) | WN (B) | MAE (B/L) | $\sigma$Reparam (B/L) |
|---|---|---|---|---|---|---|
| Top-1 | 81.8% | **82.2%** | 69.81% | 78.25% | 82.1% / 81.5% | 81.88% / **82.41%** |
| EMA Top-1 | – | – | 68.41% | 76.95% | 82.3% / 82.6% | 82.37% / 82.48% |
| Training Epochs | 300 | 300 | **250** | **250** | 300 | **250** / 300 |
| Layer Norm | Yes | **No** | **No** | Yes | Yes | **No** |
| SGD | No | No | **Yes (LARS)** | No | No | **Yes (LARS)** |
| Cosine Schedule | Yes | Yes | **No** | **No** | Yes | **No** / Yes |
| LR Warmup | Yes | Yes | **No** | **No** | Yes | **No** |
| Weight Decay | Yes | Yes | **No** | **No** | Yes | **No** |

**Why $\sigma$Reparam?**    Unlike the standard SN, $\sigma$Reparam introduces an additional multiplier $\gamma$ which explicitly controls the SN of the weights, and there is no explicit pressure to regularize the SN. The additional multiplier is necessary to avoid restricting the capacity of the network, and we find that training and overall performance is significantly degraded in its absence. Since the representational capacity of the layer remains unchanged, it is not immediately clear why $\sigma$Reparam would effectively regularize the SN of the weights. While a full theoretical characterization is beyond the scope of this paper, we identify a property of adaptive optimizers which, if left unchecked, causes the spectral norm of weight matrices to grow rapidly for large weight matrices. To illustrate this, we adopt common assumptions in stochastic optimization, and model the stochastic gradients at some point in the optimization by $g = \mu + \epsilon \in \mathbb{R}^{w \times w}$, where $\mu$ is the mean and $\epsilon$ is a random variable with $\mathbb{E}[\epsilon] = \mathbf{0}, \mathbb{E}[\epsilon^2] = n^2 \in \mathbb{R}^{w \times w}$. A typical Adam optimizer update attempts to approximate the following ideal update: $\Delta = \frac{\mathbb{E}[g]}{\sqrt{\mathbb{E}[g^2]}}$. The following proposition lower bounds the spectral norm of the ideal update $\sigma(\Delta)$:

**Proposition 3.2.** *It holds that:*

$$\sigma(\Delta) \geq \sqrt{w} \sqrt{1 - \frac{1}{w^2} \sum_{i,j=1}^{w} \frac{n_{i,j}^2}{\mu_{i,j}^2 + n_{i,j}^2}} \tag{3}$$

Note that the noise second moment $n^2$ is typically in the order of $\mu^2$, hence Eq. (3) indicates that the spectral norm of the ideal update should be large, growing linearly with $\sqrt{w}$. Moreover, for large batch sizes we would have $n^2 \ll 1$, resulting in $\sigma(\Delta) \sim \sqrt{w}$ [1]. While such a large spectral norm could be offset by a proper learning rate adjustment, this would be counter productive since 1) a small learning rate typically induces inferior performance, and 2) architectures with layers of varying sizes, such as attention layers, would require a per layer learning rate tuning. In contrast, $\sigma$Reparam avoids this issue since the spectral norm of each layer is controlled by a single parameter $\gamma$, hence the size of its update does not scale with $w$ and is uniform across layers.

## 4    EXPERIMENTS

### 4.1    SUPERVISED IMAGE CLASSIFICATION

**Improved robustness.**    We first start from a well tuned recipe with ViT-B on ImageNet-1k (Touvron et al., 2021), and vary its hyper parameters in the grid $[base\_lr \in \{5e-4, 1e-3\}, batch\_size \in \{1024, 2048\}, warmup\_epochs \in \{0, 5\}]$. 7/8 configurations lead to divergence except for the default $[5e-4, 2048, 5]$ hyper parameter. We next apply $\sigma$Reparam to all the linear layers (including the initial patch embedding), and removed all the LayerNorm instances. All configurations in the same grid search converge with an average top-1 accuracy of 81.4% (max 82.2%, shown in Table 1). This suggests improved robustness with respect to hyperparameters.

---

[1]This would be exact for full batch optimization.

**Simplified recipe.** $\sigma$Reparam also enables a simplified framework for training ViT-B and ViT-L models, in contrast to state-of-the art ImageNet-1k ViT training protocols such as the fully supervised MAE recipe (He et al., 2022) and DeiT (Touvron et al., 2021), (Table 1). In the case of ViT-B models, we are able to train for a shorter duration, remove all LayerNorm layers, remove LR warmup, remove cosine scheduling (requiring only a simple step schedule at 210 epochs) and use no weight decay. Furthermore, $\sigma$Reparam enables SGD training via LARS (You et al., 2017) (with momentum 0.9) – something not possible with traditional ViT training protocols (Touvron et al., 2021; He et al., 2022). These simplifications also have the added benefit of reducing GPU memory overhead[2]. For the ViT-L model we relax the LR schedule back to cosine and match the baseline model's training interval. Both models use FP32 precision on the attention operands and keep mixed precision training for the rest of the network. The full set of hyperparameters is available in Appendix E.

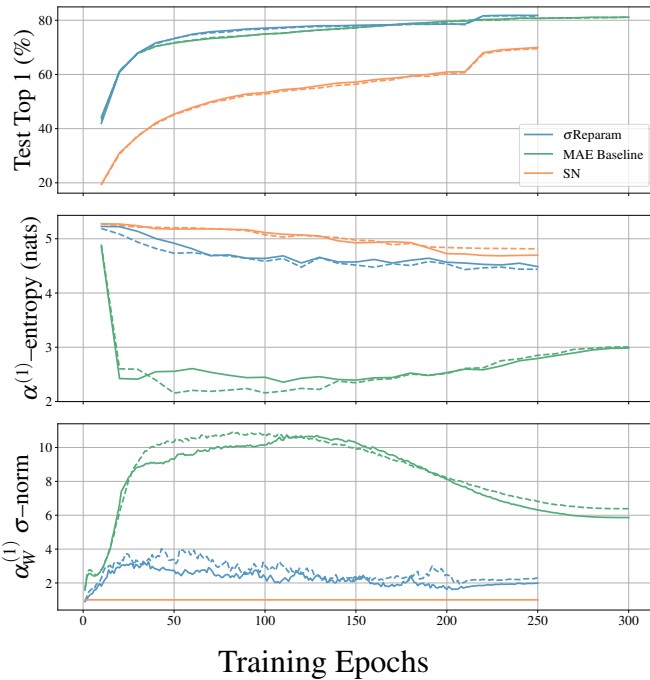

Training Epochs

Figure 2: Test performance, attention entropy, and largest singular value of attention weights of a supervised $\sigma$Reparam ViT-B/16 alongside supervised MAE ViT-B/16 and SN baselines. Best (solid line) and worst (dashed line) trials of each method are presented. The MAE ViT-B/16 presents a more constrained attention entropy in contrast to the DeiT formulation from Figure 1 due to the longer warmup, lower learning rate and stronger weight decay.

To further understand the effect of $\sigma$Reparam, we track both the attention entropy, and the largest singular value of the attention weight matrix over the course of training. In Figure 2, $\sigma$Reparam maintains a lower largest attention weight singular value and presents a higher, but monotonically decreasing attention entropy throughout training. As previously discussed, a smaller bounded singular value helps with stable training, whereas a higher attention entropy encourages exploration of more diverse solutions. This is reinforced by the accelerated performance observed in Test Top 1 and the 50 epoch reduction in training time for the $\sigma$Reparam ViT-B/16 shown in Figure 2.

## 4.2 SELF-SUPERVISED TRAINING OF VISUAL REPRESENTATIONS

In computer vision, self-supervised learning (SSL) has been effective in enabling efficient training on downstream tasks (Assran et al., 2022). Most of this progress has been made using convolutional architectures, while works using ViTs often require specialized training recipes (Caron et al., 2021).

Recently, it was found that ViTs suffer from training instabilities in SSL tasks Chen et al. (2021). These instabilities can be remedied through a combination of frozen patch embedders, initialization

---

[2]We observe a 8.2% memory reduction in full FP32 (for a 1:1 comparison) with a batch size of 86 per GPU.

Table 2: (top) Best SimCLR ImageNet1k trial top 1 linear probing performance training for 300 epochs. *σReparam + LN* yields the highest performing run, with *Frozen Patcher* performing competitively. (bottom) Configuration of the variants used in our stability analysis. The MoCo v3 weight initialization and patch initialization scheme are described in Chen et al. (2021). For full hyperparameters, see Table 6 of Appendix C.1.

|  | Baseline | Frozen Patcher | $\sigma$Reparam | $\sigma$Reparam + LN |
|---|---|---|---|---|
| Top 1 @ 300 (ours) | 72.4 | 74.4 | 73.7 | **74.5** |
| Weight Init | MoCo v3 | MoCo v3 | `trunc_norm(.02)` | `trunc_norm(.02)` |
| Patcher Init | MoCo v3 | MoCo v3 | `trunc_norm(.02)` | `trunc_norm(.02)` |
| Frozen Patcher | No | Yes | No | No |
| $\sigma$Reparam | No | No | Yes | Yes |
| Layer Norm | Yes | Yes | No | Yes |

(a) Statistics of best and worst trials per method.

(b) Stability over 10 trials per method.

Figure 3: Ten trials of SimCLR for each method on ImageNet1k with 40 epochs of learning rate warmup. **(a)** Linear probe performance for the best (solid line) and worst (dashed line) trials of each method, against relevant metrics from the first attention layer (top to bottom): attention entropy, the spectral norm of the attention weights, and the $\ell_\infty$–gradient norm of the attention weights. We see that the *Frozen Patcher* method functions as intended, regulating its gradient norm, and protecting it from the large gradient norms inducing instability in *Baseline*. We also observe a second form of instability during training: the growing spectral norm leads to a poorly behaved attention mechanism, entropy collapse, and a drop in performance as described in Section 3. This affects *Baseline*, as well as *Frozen Patcher*, as neither method gives specific protection against this second type of instability (solid and dashed red, and dashed green lines). Finally, we see that $\sigma$Reparam with and without layer normalization regulate both the gradient norms, as well as the spectral norms, giving defense against both types of instability. **(b)** Linear probe performance of every trial. We see that $\sigma$Reparam is the most stable method. $\sigma$Reparam + LN is also quite stable. In the case where it experiences instabilities, we see that it is able to recover much quicker than *Baseline* and *Frozen Patcher*. This is due to the regularization of the spectral norm which 1) prevents any arising instability pushing the model too far away from the current solution, and 2) keeps the attention mechanism useful, such that gradients are available for any required correction.

schemes, and longer learning rate warmups; however, there is an open question whether a general solution providing stable SSL ViT training exists Chen et al. (2021).

Here, we demonstrate that $\sigma$Reparam is a ViT SSL stabilizer. Taking SimCLR as our SSL method, we investigate four variants. *Baseline* and *Frozen Patcher* were studied in Chen et al. (2021), whereas *σReparam* and *σReparam + LN* are our solution. These methods are detailed in Table 2 and their full hyperparameters given in Table 6 of Appendix C.1.

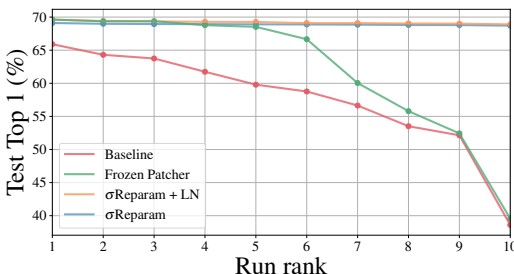

Figure 4: Linear probe performance on ImageNet1k at the end of training over 10 trials for each method. Trials are ordered by decreasing performance, with run rank 1 (10) corresponding to the best (worst) trial. *Frozen Patcher* and *σReparam + LN* produce the best individual runs, with *σReparam* marginally lower. *σReparam + LN* and *σReparam* are the methods most reliably giving good performance, with *Baseline* and *Frozen Patcher* each susceptible to at least one instability type.

We observe two types of instability. The first, as observed in Chen et al. (2021), is induced by large gradient norms in early layers. The second, described in Section 3, relates to entropy collapse. We find that *Frozen Patcher* protects against the first type, but is still susceptible to the second. σReparam, however, can protect against both types of instability, yielding more reliable training (see Figure 3).

As noted in Chen et al. (2021), instabilities reduce final performance. We show instability impact on performance in Figure 4. The methods with the best performing individual runs are *Frozen Patcher* and *σReparam + LN*, whereas the most stable methods are *σReparam + LN* and *σReparam*.

Our main stability experiments use 40 epochs of learning rate warmup, matching the setting of Chen et al. (2021). Using σReparam, as in the supervised setting, gives training stability even at the lower learning rate warmup of 10 epochs. For more details, see Appendix C.2.

Finally, we look at the performance attainable when training for a longer duration of 300 epochs in Table 2. The best performing method run is given by with *σReparam + LN*, with *Frozen Patcher* performing almost as well, and both outperforming the reference SimCLR result (Chen et al., 2021).

Ultimately, we see while *σReparam* produces the lowest degree of instability, the best overall method for stable training of SimCLR ViTs is *σReparam + LN*, producing both the highest ImageNet1k linear probe performance at 100 epochs (69.6 %) and 300 epochs (74.5 %) epochs, as well as very stable training over many trails, both at long and short learning rate warmup.

## 4.3 SPEECH

In this section we focus on experiments for automatic speech recognition (ASR).

**Data**  All experiments are performed on the subset of 100h audio paired with transcriptions (*train-clean-100*) of LibriSpeech dataset Panayotov et al. (2015). The standard LibriSpeech validation sets (*dev-clean* and *dev-other*) are used to tune all hyper parameters, as well as to select the best models. Test sets (*test-clean* and *test-other*) are used only to report final word error rate (WER) performance without an external language model. We keep the original 16kHz sampling rate and compute log-mel filterbanks with 80 coefficients for a 25ms sliding window, strided by 10ms, later normalized to zero mean and unit variance per input sequence.

**Acoustic Models**  We are using current, to the best of our knowledge, state-of-the-art model on 100h of LibriSpeech (Likhomanenko et al., 2021a). The model consists of 1D convolution to perform striding, Transformer encoder with post-LN and a final linear layer to map to the output number of tokens[3]. The model is trained with Connectionist Temporal Classification (Graves et al., 2006) loss. To speed up the model training (2-3x) and decrease memory usage we are using CAPE positional embeddings (Likhomanenko et al., 2021c) instead of relative embeddings Shaw et al. (2018).

**Data augmentation**  We use SpecAugment (Park et al., 2019) activated right at the beginning of training. We use two frequency masks with frequency mask parameter $F = 30$, ten time masks with maximum time-mask ratio $p = 0.1$ and time mask parameter $T = 50$; time warping is not used.

**Training**  We use Adagrad (Duchi et al., 2011) if not specified otherwise, and LR decaying by 2 each time the WER reaches a plateau on the validation. We use dynamic batching of 240s audio per GPU and train with tensor cores fp32 on 8 Ampere A100 (40GB) GPUs for 350-500k updates. No

---

[3]The token set consists of the 26 English alphabet letters augmented with the apostrophe and a word boundary token.

Table 3: Comparison between different normalizations and our re-parametrization for speech domain: training loss and word error rate are reported for the best models.

|  | post-LN | pre-LN (same) | pre-LN (optimized) | SN | SN +post-LN | WN | WN +post-LN | $\sigma$Reparam | $\sigma$Reparam +post-LN |
|---|---|---|---|---|---|---|---|---|---|
| Train Loss | 37.7 | 35.3 | 37.2 | 160.4 | 120.3 | 35.6 | 35.4 | 37.5 | 34.9 |
| dev-clean WER | 5.9 | 6.9 | 6.2 | 42.6 | 20.3 | 7.0 | 6.3 | 6.4 | 6.1 |
| dev-other WER | 17.7 | 21.3 | 19.1 | 62.9 | 42.7 | 22.3 | 19.4 | 20.5 | 17.8 |
| test-clean WER | 6.2 | 7.1 | 6.3 | 42.4 | 20.4 | 7.3 | 6.7 | 6.8 | 6.4 |
| test-other WER | 17.8 | 21.6 | 19.3 | 63.6 | 43.6 | 22.6 | 19.5 | 21.0 | 18.0 |

weight decay is used. Default warmup is set to 64k for the baselines and varied for different models. The default LR is 0.03 and also optimized across models. We also apply gradient clipping of 1.

### 4.3.1 TRAINING STABILITY, ROBUSTNESS AND GENERALIZATION

First, we experiment with stability of training for the baselines using both "Pre Norm" (pre-LN) and "Post Norm" (post-LN) architectures. If we vary LR, warmup, and gradient clipping, all post-LN experiments either diverge or no training is observed. At the same time, pre-LN is stable: we can reduce warmup from 64k to 16k, increase learning rate from 0.03 to 0.5, and obtain better results than before. While pre-LN is more stable than post-LN, it generalizes worse: validation WER is worse while training loss is lower, see Table 3. When we switch to $\sigma$Reparam we observe the same stability as for pre-LN, Figure 5, while having better generalization than not optimized pre-LN. We are not able to match the post-LN results until we combine post-LN together with $\sigma$Reparam, which allows us to achieve similar performance on the dev and test sets and lower training loss. In Figure 5 both $\sigma$Reparam and $\sigma$Reparam with post-LN demonstrate robustness with respect to training hyperparameters. We also compare with Spectral Norm (SN) where $\gamma$ is set to 1 and is not learnable and WN baselines. Both SN and WN perform poor compared to $\sigma$Reparam, see Table 3.

In prior works it was reported that post-LN can be impossible to train with very deep architectures, see e.g. Liu et al. (2020b;a). We reproduced similar results: if we increase the encoder size to 2x then post-LN does not train, while pre-LN works out of the box and improves over the smaller architecture. We applied the same settings to $\sigma$Reparam and combination of $\sigma$Reparm and post-LN: for both cases out of the box models train well and achieve similar results as pre-Norm. This confirms $\sigma$Reparam's ability for stable training even with post-LN.

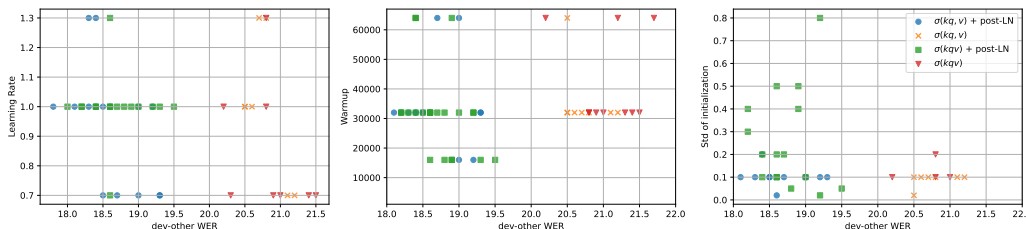

Figure 5: Robustness of $\sigma$Reparam with respect to different hyperparameters: learning rate (left), warmup (middle) and initialization std value (right).

### 4.3.2 TRAINING WITH SGD

Prior works report different problems training transformers with SGD (see e.g. (Li et al., 2022)). First, we experimented with the baselines, pre-LN and post-LN and observed similar issues. It is hard to find hyperparameters that enable the model to train. Following vision experiments we switch to the LARS (You et al., 2017) (with momentum 0.9) optimizer, and are able to train pre-LN and post-LN by carefully tuning the LR (the rest stays the same, including gradient clipping) which is varied from 0.1 to 1.5, see Table 4. First, we observe that post-LN is more unstable (many LRs are diverging or not training) and gives significantly worse results than pre-LN. Second, pre-LN is still behind the baseline that uses an adaptive optimizer. However, if we switch to $\sigma$Reparam (key, queries and values are represented as one matrix) we observe stable training with respect to LR changes,

Table 4: Comparison between different normalizations and our re-parametrization for speech domain when no warmup and LARS optimizer are used: training loss and word error rate are reported for the best models. $\sigma$Reparam performs re-parametrization for joint matrix for key, queries and values in self-attention. DV denotes model divergence: we are not able to train SN with post-LN configuration.

|  | post-LN | pre-LN | SN | SN +post-LN | WN | WN +post-LN | $\sigma$Reparam | $\sigma$Reparam +post-LN |
|---|---|---|---|---|---|---|---|---|
| Train Loss | 64.5 | 29.4 | 160.0 | DV | 59.1 | 63.2 | 51.1 | 34.2 |
| dev-clean WER | 8.1 | 5.9 | 49.8 | DV | 8.3 | 7.1 | 7.2 | 5.8 |
| dev-other WER | 25.0 | 18.9 | 69.6 | DV | 25.9 | 22.0 | 22.8 | 18.1 |
| test-clean WER | 8.6 | 6.4 | 49.4 | DV | 8.7 | 7.5 | 7.5 | 6.2 |
| test-other WER | 25.6 | 19.2 | 70.9 | DV | 26.4 | 22.1 | 23.2 | 18.7 |

and combined together with post-LN it achieves similar performance to the best results from Table 3 while keeping the train loss low[4].

## 4.4 LANGUAGE

**Setup.** We use the WikiText-103 language model (LM) benchmark, which consists of 103M tokens sampled from English Wikipedia (Merity et al., 2017). Our baseline is a highly optimized Transformer (Baevski & Auli, 2019) with 32 layers, 8 heads, 128 head dimensions, 1024 model dimensions, 4096 fully connected dimensions and post LayerNorm. The word embedding and softmax matrices are tied (Press & Wolf, 2017). We partition the training data into non-overlapping blocks of 512 contiguous tokens and train the model to autoregressively predict each token (Baevski & Auli, 2019). Validation and test perplexity is measured by predicting the last 256 words out of the input of 512 consecutive words to avoid evaluating tokens in the beginning with limited context (*early token curse*, Press et al., 2021).

Table 5: WikiText-103 language modeling results in perplexity.

| Model | PPL↓ | | |
|---|---|---|---|
| | train | dev. | test |
| $\sigma$Reparam w/ weight decay | 16.5 | **17.9** | **18.6** |
| $\sigma$Reparam w/o weight decay | **12.9** | 18.5 | 19.3 |
| Baseline Transformer Baevski & Auli (2019) | 15.4 | 18.1 | 18.7 |

**Results.** We do not experience training instability with the baseline Transformer, likely because the masked attention in autoregressive models makes entropy collapse less likely to occur. Nonetheless, we experimented with $\sigma$Reparam to test its generality on a different modality/problem. We apply $\sigma$Reparam to all linear layers of the Transformer while removing all LayerNorms, and search for learning rate in a grid [1, 1.5, 2, 2.5] and weight decay in the grid [1e-3, 1e-4, 0]. All other hyperparameters are kept the same as the baseline. The results are shown in Table 5. We see that even in the absence of LayerNorm, $\sigma$Reparam shows strong performance in convergence and dev/test performance. With a mild weight decay, $\sigma$Reparam also outperforms the baseline wrt the dev/test PPL.

## 5 CONCLUSION

We analyze the training stability of Transformers from the lens of the attention entropy. We show that training instability or divergence is often accompanied by the entropy collapse phenomenon, and provide a simple fix named $\sigma$Reparam. We demonstrate over a wide set of benchmarks, domains, and training methodologies, that $\sigma$Reparam provides great stability and robustness, often leading to simplified model design and/or better performance.

---

[4]For the separate reparametrization for (keys, queries) and values we observe less stable training with LARS and no warmup relative to reparametrizing them together.

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

# A PROOF OF THEOREM 3.1 AND PROPOSITION 3.2

**Theorem 3.1** (Attention entropy lower bound). *Assume without loss of generality $\|X\|_2 \leq 1$, and let spectral norm $\sigma = \|W_K W_Q^\top\|_2$. Then it holds that:*

$$Ent(A_i) \geq \log\left(1 + (T-1)e^{-\sigma\sqrt{\frac{T}{T-1}}}\right) + \frac{\sigma\sqrt{T(T-1)}e^{-\sigma\sqrt{\frac{T}{T-1}}}}{1 + (T-1)e^{-\sigma\sqrt{\frac{T}{T-1}}}} \tag{1}$$

*Moreover, there exists inputs $X$ and weights $W_K, W_Q$ for which the lower bound in Eq. (1) is tight.*

*Proof.* WLOG let $u \in \mathbb{R}^T$ denote the $j$'th row of $a$. From the condition that $\|X\|_2 \leq 1$ it holds that $\|u\| \leq \sigma$. Let $p = p(u)$ denote the softmax probabilities given by:

$$p_i = \frac{e^{u_i}}{Z} \tag{4}$$

where $Z = \sum_{j=1}^T e^{u_j}$ is the partition function. The entropy given $p(u)$ is then:

$$\text{Ent}(u) = -\sum_{i=1}^T \frac{e^{u_i}}{Z}\log(\frac{e^{u_i}}{Z}) = -\sum_{i=1}^T \frac{u_i e^{u_i}}{Z} + \log(Z). \tag{5}$$

We wish to solve the following constrained minimization problem:

$$\min_u \text{Ent}(u) \ \text{ s.t } \ \|u\|^2 \leq \sigma^2 \tag{6}$$

where $D > 0$. Define the lagrangian:

$$\mathcal{L}(u, \lambda) = \text{Ent}(u) + \frac{1}{2}\lambda(\|u\|^2 - \sigma^2) \tag{7}$$

To find all saddle points, we solve the system of equations:

$$\frac{\partial \mathcal{L}(u, \lambda)}{\partial u} = 0, \quad \frac{\partial \mathcal{L}(u, \lambda)}{\partial \lambda} = 0 \tag{8}$$

Giving rise to the following set of equations:

$$\forall_{1 \leq k \leq T}, \ \lambda u_k = \sum_{i=1}^T \frac{e^{u_i}}{Z}(\delta_{i,k} - \frac{e^{u_k}}{Z})(1 + \log(\frac{e^{u_i}}{Z})) \tag{9}$$

$$= p_k(\log(p_k) + \text{Ent}(u)) \tag{10}$$

$$\|u\|^2 = \sigma^2 \tag{11}$$

As a first step, assume that for the minimizer $u^\star$ of Eq. (6) there exists an index $k$ such that $u_k^\star = 0$. Using Eq. (7):

$$0 = \log(p_k) + \text{Ent}(u) = -\sum_{i=1}^T p_i \log(\frac{p_i}{p_k}) = -\sum_{i=1}^T p_i \log(e^{u_i}) = -\sum_{i=1}^T p_i u_i = -\mathbb{E}u \tag{12}$$

From the first set of equations we arrive at the condition:

$$\forall_{u_j, u_{j'} \neq 0}, \ p_j \frac{\log(p_j) + \text{Ent}(u)}{u_j} = p_{j'} \frac{\log(p_{j'}) + \text{Ent}(u)}{u_{j'}} \tag{13}$$

$$\longrightarrow p_j + \frac{\mathbb{E}u}{u_j} = p_{j'} + \frac{\mathbb{E}u}{u_{j'}} \tag{14}$$

$$\longrightarrow p_j = p_{j'} \tag{15}$$

This however implies that $u_1^\star = u_2^\star = ... = u_T^\star = 0$, hence a contradiction to Eq. (9). Now, assuming $\forall_k \ u_k \neq 0$, we have that:

$$\forall_{u_j \neq u_{j'}} \frac{e^{u_j} - e^{u_{j'}}}{\frac{1}{u_{j'}} - \frac{1}{u_j}} = Z\mathbb{E}u = \text{const} \tag{16}$$

The monotonicity of the LHS of Eq. (16) implies that $u$ contains only 2 distinct values. WLOG assume $u_1^\star = \alpha, \forall_{i>1}, u_i^\star = -\sqrt{\frac{D^2-\alpha^2}{T-1}}$. Then we have:

$$\frac{e^\alpha - e^{-\sqrt{\frac{\sigma^2-\alpha^2}{T-1}}}}{-\frac{1}{\sqrt{\frac{\sigma^2-\alpha^2}{T-1}}} - \frac{1}{\alpha}} = \alpha e^\alpha + (1-T)\sqrt{\frac{\sigma^2-\alpha^2}{1-T}}e^{-\sqrt{\frac{\sigma^2-\alpha^2}{T-1}}} \tag{17}$$

With a solution:

$$\alpha = \sigma\sqrt{1-\frac{1}{T}}, \quad \beta = -\sigma\sqrt{\frac{1}{T(T-1)}} \tag{18}$$

With the corresponding entropy:

$$\text{Ent}(u^\star) = \log\left(1 + (T-1)e^{-\sigma\sqrt{\frac{T}{T-1}}}\right) + \frac{\sigma\sqrt{T(T-1)}e^{-\sigma\sqrt{\frac{T}{T-1}}}}{1 + (T-1)e^{-\sigma\sqrt{\frac{T}{T-1}}}} \tag{19}$$

$\square$

**Proposition A.1.** *It holds that:*

$$\sigma(\Delta) \geq \sqrt{w}\sqrt{1 - \frac{1}{w^2}\sum_{i,j=1}^{w}\frac{n_{i,j}^2}{\mu_{i,j}^2 + n_{i,j}^2}} \tag{3}$$

*Proof.* We have that:

$$\sigma(\Delta) \geq \frac{1}{\sqrt{w}}\sqrt{\text{Trace}(\Delta^\top\Delta)} = \frac{1}{\sqrt{w}}\sqrt{\sum_{i,j=1}^{w}\frac{\mu_{i,j}^2}{\mu_{i,j}^2 + n_{i,j}^2}} = \sqrt{w}\sqrt{1 - \frac{1}{w^2}\sum_{i,j=1}^{w}\frac{n_{i,j}^2}{\mu_{i,j}^2 + n_{i,j}^2}} \tag{20}$$

$\square$

## B  IMPLEMENTATION OF $\sigma$REPARAM

To compute spectral norm of the current matrix we use the power method as approximation method to speed up computations. See Algorithm 1 for a sketch implementation.

---

**Algorithm 1** Pseudo code of $\sigma$Reparam in a PyTorch-like style.

---

```
# parameters. W: weight matrix, shape (d, c); gamma: the learned spectral norm, shape (1,)
# buffers. u: shape (d,), v: shape (c,), the left and right singular vectors of W
if init: # initialize u, v as random unit vectors and gamma to 1
    u = randn(d)
    u = u / u.norm(dim=0)
    v = randn(c)
    v = v / v.norm(dim=0)
    gamma = ones(1)
if training: # if in the training mode, perform one step of power iteration first
    u = W.mv(v)
    u = u / u.norm(dim=0)
    v = W.T.mv(u)
    v = v / v.norm(dim=0)
sigma = einsum('d,dc,c->', u, W, v)
W_hat = gamma / sigma * W # the effective spectral norm of W_hat would be gamma
```

---

Table 6: Default hyperparameters of the variants of SimCLR used in our stability analysis. The MoCo v3 weight initialization and patch initialization scheme are described in Chen et al. (2021). SinCos refers to stacked 2D SinCos positional encodings Vaswani et al. (2017). The table is divided vertically into hyperparameters that differ across methods (top) and hyperparameters shared across methods (bottom).

| | Baseline | Frozen Patcher | $\sigma$Reparam | $\sigma$Reparam + LN |
|---|---|---|---|---|
| $\sigma$Reparam | No | No | Yes | Yes |
| Frozen Patcher | No | Yes | No | No |
| Layer Norm | Yes | Yes | No | Yes |
| Patcher Init | MoCo v3 | MoCo v3 | `trunc_norm(.02)` | `trunc_norm(.02)` |
| Weight Init | MoCo v3 | MoCo v3 | `trunc_norm(.02)` | `trunc_norm(.02)` |
| Architecture | ViT-B/16 | ViT-B/16 | ViT-B/16 | ViT-B/16 |
| Batch Size | 4096 | 4096 | 4096 | 4096 |
| ColorJitter Strength | 0.5 | 0.5 | 0.5 | 0.5 |
| Learning Rate | $2 \times 10^{-4}$ | $2 \times 10^{-4}$ | $2 \times 10^{-4}$ | $2 \times 10^{-4}$ |
| Learning Rate Sched | Cosine | Cosine | Cosine | Cosine |
| Learning Rate Warmup | 40 Epochs | 40 Epochs | 40 Epochs | 40 Epochs |
| Optimizer | AdamW | AdamW | AdamW | AdamW |
| Positional Encoding | SinCos | SinCos | SinCos | SinCos |
| Weight Decay | 0.1 | 0.1 | 0.1 | 0.1 |

## C  SELF-SUPERVISED TRAINING OF VISUAL REPRESENTATIONS

### C.1  HYPERPARAMETERS

Here we outline the hyperparameters of our experimental setup for SimCLR+ViT stability. For the variations, alongside their default hyperparameters see Table 6. These hyperparameters are used in all SimCLR runs unless stated otherwise.

**Augmentations**  We use SimCLR augmentations throughout, however, we run at half ColorJitter strength, equal to the ColorJitter strength of MoCo v3. For completeness, we provide our training augmentation here, our testing augmentation is the standard resize, center crop and normalize. Half color strength corresponds to `color_jitter_strength = 0.5`. Setting `color_jitter_strength = 1.0` recovers the base SimCLR training augmentations.

```
[
    transforms.RandomResizedCrop(
        image_size_override, scale=crop_scale, interpolation=Image.BICUBIC
    ),
    transforms.RandomHorizontalFlip(p=0.5),
    transforms.RandomApply(
        [
            transforms.ColorJitter(
                brightness=0.8 * color_jitter_strength,
                contrast=0.8 * color_jitter_strength,
                saturation=0.8 * color_jitter_strength,
                hue=0.2 * color_jitter_strength,
            )
        ],
        p=0.8,
    ),
    transforms.RandomGrayscale(p=0.2),
    transforms.RandomApply([M.GaussianBlur([0.1, 2.0])], p=0.5),
    transforms.ToTensor(),
    IMAGENET_NORMALIZE,
]
```

### C.2  REDUCED LEARNING RATE WARMUP

In Chen et al. (2021) the authors noted that the learning rate warmup period needed extending from its typical ImageNet1k default of 10 epochs to 40 epochs, enhancing the stability of the method. We observe that using $\sigma$Reparam, either with or without Layer Norm, we are able to achieve stable SimCLR+ViT training at the original warmup period of 10 epochs (see Figure 6).

As with our analysis at the longer warmup period, we also investigate the performance distribution across the trials, giving a sense of how instability impacts the final model (see Figure 6).

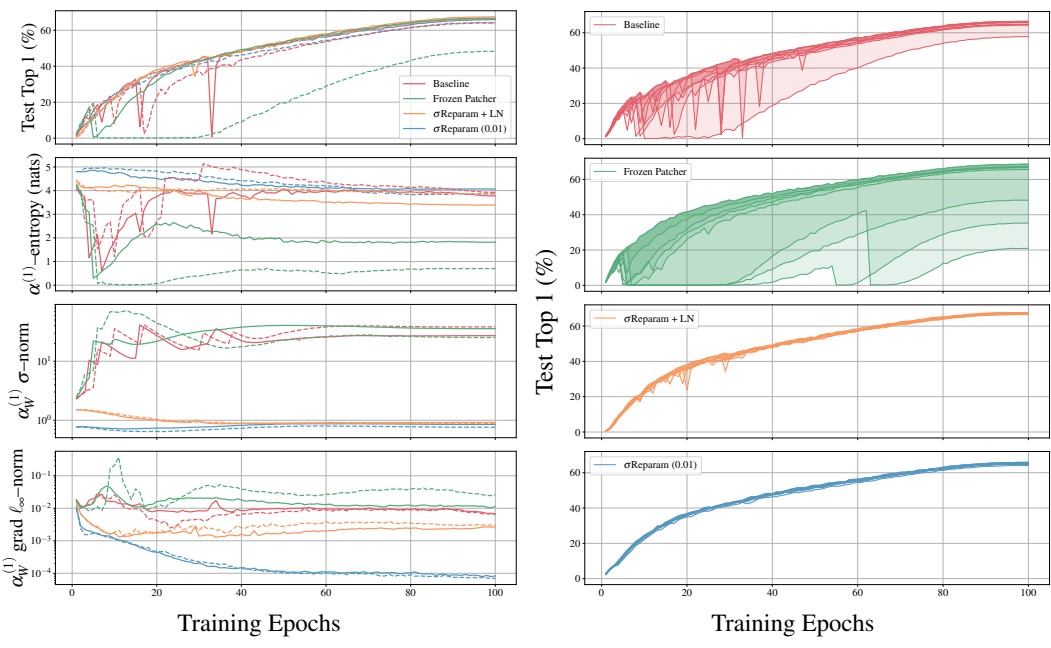

(a) Statistics of best and worst trials per method.

(b) Stability over 8 trials per method.

Figure 6: Eight trials of SimCLR for each method on ImageNet1k with 10 epochs of learning rate warmup. **(a)** Linear probe performance for the best (solid line) and worst (dshed line) trials of each method, against relevant metrics from the first attention layer (top to bottom): attention entropy, the spectral norm of the attention weights, and the $\ell_\infty$−gradient norm of the attention weights. Our observations are consistent with those of the longer warmup of 40 epochs investigated in Figure 3, except that here, *Frozen Patcher* is less able to tame early layer gradient norms than it was in the longer warmup (dashed green line). **(b)** Linear probe performance of every trial. Observations are again consistent with the longer warmup; $\sigma$Reparam with and without Layer Norm are the most stable methods. *$\sigma$Reparam (0.01)* refers to a $\sigma$Reparam with an initialization scheme of `trunc_normal(.01)` instead of `trunc_normal(.02)`, with the former showing some signs of instability. Understanding the source of this instability will be the subject of future work. *$\sigma$Reparam + LN* uses the default `trunc_normal(.02)`.

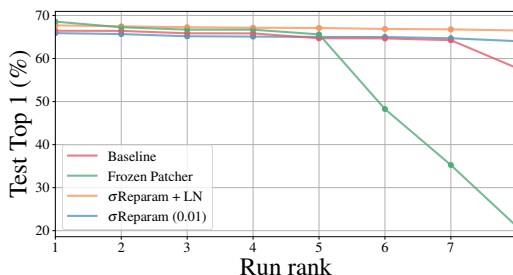

Figure 7: Linear probe performance on ImageNet1k at the end of training over 8 trials for each method. Trials are ordered by decreasing performance, with run rank 1 (8) corresponding to the best (worst) trial. *Frozen Patcher* produce the best individual, with all other methods marginally lower. *$\sigma$Reparam + LN* and *$\sigma$Reparam* are the methods most reliably giving good performance, with *Baseline* and *Frozen Patcher* each susceptible to at least one instability type.

# D  SPEECH EXPERIMENTS

## D.1  ABLATIONS ON THE INITIALIZATION FOR $\sigma$REPARAM

First, we found that it is better to initialize $\gamma$ as 1 and not compute it from the initialized kernel as there could be different values for spectral norm depending on the initialization of the kernel. In this case we observed values greater than 1 for the spectral norm which cause divergence / no training. From practical point it is native to keep $\gamma = 1$. We compared different initializations for kernel and we didn't see any differences in initialization (e.g. uniform, normal). The only thing influences is the std of the initialization pdf which influences also effective LR. In speech we found that training is robust with respect to changes of std (Figure 5), however larger std performs better and sweet spot is 0.2-0.3.

## D.2 Full LibriSpeech Experiments

We also evaluate $\sigma$Reparam for large scale data in speech domain: we take now the whole LibriSpeech as the training data. We consider again Adagrad optimizer with two schedules on learning rate: cosine (with 1 phase of 500k iterations) and step-wise decaying as before for *train-clean-100* experiments. We use exactly the same architecture and hyper-parameters as in Table 9 except dropout and layer drop which are decreased to 0.1 to decrease model regularization. For all models we tune only learning rate. Keys and queries spectral reparametrization is done separately from values, also we use learning rate on gamma to be twice bigger than the main learning rate. Our experiments as for *train-clean-100* show, see Tables 7 and 8, that $\sigma$Reparam accompanied with post-LN can match the post-LN baseline, while having robustness to the hyper-parameter changes (e.g. allows larger learning rate values without any issues).

Table 7: Comparison between different normalizations and our re-parametrization for speech domain on full LibriSpeech with step-wise LR schedule: word error rate are reported for the best models.

|  | post-LN (Likhomanenko et al., 2021b) | post-LN | pre-LN (same) | pre-LN (optimized) | $\sigma$Reparam | $\sigma$Reparam +post-LN |
|---|---|---|---|---|---|---|
| dev-clean WER | 2.6 | 2.6 | 2.9 | 2.6 | 2.7 | 2.8 |
| dev-other WER | 7.0 | 6.9 | 7.7 | 6.8 | 7.2 | 7.1 |
| test-clean WER | 2.7 | 2.7 | 3.0 | 2.8 | 2.9 | 2.9 |
| test-other WER | 6.9 | 6.9 | 7.8 | 6.8 | 7.3 | 7.0 |

Table 8: Comparison between different normalizations and our re-parametrization for speech domain on full LibriSpeech with cosine LR schedule: word error rate are reported for the best models.

|  | post-LN | pre-LN (same) | $\sigma$Reparam | $\sigma$Reparam +post-LN |
|---|---|---|---|---|
| dev-clean WER | 2.6 | 2.6 | 2.8 | 2.7 |
| dev-other WER | 7.1 | 6.9 | 7.6 | 7.3 |
| test-clean WER | 2.9 | 2.8 | 3.0 | 2.9 |
| test-other WER | 7.2 | 7.0 | 7.7 | 7.2 |

## D.3 Ablations on separate $\sigma$Reparam for key, queries and values

We found that in the end they behaves more or less similar while separate normalization allows to achieve lower training loss due to larger capacity ability which provides potential to scale. However, for training with LARS it is better to have joint re-parametrization to achieve stable training and comparable results with adaptive optimizers, see Section 4.3.2.

## D.4 Hyperparameters

We present hyperparameters for our speech experiments in Table 9 and speech experiments with LARS in Table 10.

Table 9: Training hyperparameter comparison for speech domain, Table 3.

| | post-LN | pre-LN | $\sigma$Reparam | $\sigma$Reparam + post-LN |
|---|---|---|---|---|
| dev-clean | 5.9 | 6.2 | 6.4 | 6.1 |
| dev-other | 17.7 | 19.1 | 20.5 | 17.8 |
| Weight Init | uniform(.036) | uniform(.036) | trunc_normal(.1) | trunc_normal(.1) |
| $\sigma$Reparam | No | No | Yes | Yes |
| Layer Norm | Yes | Yes | No | Yes |
| Base LR | 0.03 | 0.5 | 1 | 1 |
| Optimizer | Adagrad | | | |
| LR schedule | step(330k, 0.5) | | | |
| Batch size | 240s x 8 | | | |
| Weight decay | none | | | |
| Warmup steps | 64k | | | |
| Training steps | 500k | | | |
| Dropout | 0.3 | | | |
| Stoch. Depth | 0.3 | | | |
| SpecAugment | $F = 30, T = 50, p = 0.1, fmask = 2, tmask = 10$ | | | |
| Grad. clip | 1 | | | |

Table 10: Training hyperparameter comparison for speech domain trained with LARS, Table 4.

| | post-LN | pre-LN | $\sigma$Reparam | $\sigma$Reparam + post-LN |
|---|---|---|---|---|
| dev-clean | 8.1 | 5.9 | 7.2 | 5.8 |
| dev-other | 25 | 18.9 | 22.8 | 18.1 |
| Weight Init | uniform(.036) | uniform(.036) | trunc_normal(.1) | trunc_normal(.1) |
| $\sigma$Reparam | No | No | Yes | Yes |
| Layer Norm | Yes | Yes | No | Yes |
| Base LR | 0.1 | 0.5 | 1 | 0.3 |
| Optimizer | LARS | | | |
| Momentum | 0.9 | | | |
| LR schedule | step(300k, 0.5) | | | |
| Batch size | 240s x 8 | | | |
| Weight decay | none | | | |
| Warmup steps | 0k | | | |
| Training steps | 500k | | | |
| Dropout | 0.3 | | | |
| Stoch. Depth | 0.3 | | | |
| SpecAugment | $F = 30, T = 50, p = 0.1, fmask = 2, tmask = 10$ | | | |
| Grad. clip | 1 | | | |

# E   HYPERPARAMETERS FOR SUPERVISED VISION

As mentioned in Section 4.1 we compare $\sigma$Reparam against DeiT (Touvron et al., 2021) and the MAE (He et al., 2022) (Appendix A.2) supervised training recipes for vision transformers. In Table 11 we highlight the differences between DeiT, MAE supervised and $\sigma$Reparam. $\sigma$Reparam presents a simplified and stable training objective for ViT-B variants. In Table 12 we present the same comparing the ViT-L variants. There is no exact 1:1 comparison for a ViT-L with the DeiT training framework so we only compare against the MAE supervised model.

Table 11: Training hyper-parameter comparison for supervised ViT-B/16.

|  | DeiT | MAE | $\sigma$Reparam |
|---|---|---|---|
| Top-1 | 81.8% | 82.1% | 81.88% |
| EMA Top-1 | - | 82.3% | 82.37% |
| Weight Init | trunc_normal(.02) | trunc_normal(.02) | trunc_normal(.02) |
| Patcher Init | trunc_normal(.02) | trunc_normal(.02) | trunc_normal(.02) |
| $\sigma$Reparam | No | No | Yes |
| Layer Norm | Yes | Yes | **No** |
| Optimizer | AdamW($\beta_1$=0.9, $\beta_2$=0.95) | AdamW($\beta_1$=0.9, $\beta_2$=0.95) | **LARS(mom=0.9)** |
| Base LR | $5 \times 10^{-4}$ | $1 \times 10^{-4}$ | 0.1 |
| LR schedule | cosine | cosine | **step(210, 0.1)** |
| Batch size | 1024 | 4096 | 4096 |
| Weight decay | 0.05 | 0.3 | **0.0** |
| Warmup epochs | 5 | 20 | **0** |
| Training epochs | 300 | 300 | **250** |
| Label smoothing | 0.1 | 0.1 | 0.1 |
| Stoch. Depth | 0.1 | 0.1 | 0.1 |
| Repeated Aug. | 2 | 2 | 2 |
| RandAug | 9/0.5 | 9/0.5 | 9/0.5 |
| Mixup prob. | 0.8 | 0.8 | 0.8 |
| Cutmix prob. | 1.0 | 1.0 | 1.0 |
| Erasing prob. | 0.25 | 0.25 | 0.25 |

Table 12: Training hyperparameter comparison for supervised ViT-L/16.

|  | MAE | $\sigma$Reparam |
|---|---|---|
| Top-1 | 81.5% | 82.41% |
| EMA Top-1 | 82.6% | 82.48% |
| Weight Init | trunc_normal(.02) | trunc_normal(.01) |
| Patcher Init | trunc_normal(.02) | trunc_normal(.0025) |
| $\sigma$Reparam | No | Yes |
| Layer Norm | Yes | **No** |
| Optimizer | AdamW($\beta_1$=0.9, $\beta_2$=0.95) | **LARS(mom=0.9)** |
| Base LR | $1 \times 10^{-4}$ | 0.15 |
| LR schedule | cosine | cosine |
| Batch size | 4096 | 4096 |
| Weight decay | 0.3 | **0.0** |
| Warmup epochs | 20 | **0** |
| Training epochs | 300 | 300 |
| Label smoothing | 0.1 | 0.1 |
| Stoch. Depth | 0.2 | 0.2 |
| Repeated Aug. | 2 | 2 |
| RandAug | 9/0.5 | 9/0.5 |
| Mixup prob. | 0.8 | 0.8 |
| Cutmix prob. | 1.0 | 1.0 |
| Erasing prob. | 0.25 | 0.25 |

