# OpenReview forum: "$\sigma$Reparam: Stable Transformer Training with Spectral Reparametrization"
_ICLR.cc/2023/Conference — Submitted to ICLR 2023_

### Official Review · Reviewer_qd4k · 2022-10-24

**Confidence:** 4
**Clarity, Quality, Novelty And Reproducibility:** Please refer to the weaknesses section.
**Correctness:** 3
**Technical Novelty And Significance:** 2
**Empirical Novelty And Significance:** 2
**Recommendation:** 3

**Strength And Weaknesses:**

Strength:
The main reason to accept this paper is empirical results and showing performance o various tasks. The author has provided more quantitative results on various tasks.


Weaknesses:

The current literature survey does not discuss the advantages and disadvantages of the paper. The author should validate why I should include this paper in this current work.

How did you get this equation-2? Is there any intuition of this equation?


The generalization of the method. The equation-2 is valid for the linear layer. What happened to the convolution layer? Is this valuable reparametrization for the convolution network?

The paper does not explain the relation between attention entropy and \sigma_reparameterisation.

How does this learnable parameter contribute to the stability of the transformer?

Instead of evaluating the proposed method on multiple tasks, the author could have done an ablation analysis on different kinds of networks and various types of transformer architecture. The author should provide ablation analysis results to clarify the proposed method's work principle.

Why do we need such reparametrization, learning another parameter? Did you analyze other kinds of techniques to tackle the attention entropy collapse problem? This requires a substantial literature survey and motivation for the proposed method. Can you justify this? How is the attention entropy collapse solved by the proposed reparametrization technique?


The author should compare the stability analysis results of this transformer with the existing state-of-the-art stable transformer. Also need to add a small paragraph in the related work section about stability/robustness in the transformers.

Can you elaborate on the number of parameters and FLOPS required for this? What about the complexity of the proposed method?

The name spectral reparametrization in the title makes it confusing here. It seems it works on frequency domain reparametrization.


**Summary Of The Paper:**

The author has discussed the transformers models' training stability on various tasks. The author has investigated the sharpness of attention concerning attention entropy on each attention head during training. The author observes that the attention entropy first decreases then increases, and then enters into the long stable stage. To address the issue, the author has proposed a simple technique \sigma_reparametrization, which reparametrize all the linear layers with spectral normalization and learnable scalar multiplication. Finally, the author has evaluated the proposed techniques on several tasks, such as supervised image classification, self-supervised learning, automatic speech recognition, and language modeling tasks, and claimed better stability and robustness concerning the choice of hyperparameters.

**Summary Of The Review:**


Justification :
The overall writing quality is ok, and the proposed method is simple and beneficial. However, the experiment comparison lacks justification, and the technical contribution is plain.

---

> ### Author Response · Authors · 2022-11-18
> **Response 1/2**
>
> Dear Reviewer qd4k,
>
> Thanks for your time, valuable feedback and acknowledgment of our work. Please find below our responses to your comments and suggestions:
>
>
> > The current literature survey does not discuss the advantages and disadvantages of the paper. The author should validate why I should include this paper in this current work.
>
>
> In order to justify the contribution of this paper, we need to first acknowledge that state of the art Transformers have poor robustness wrt training hyper parameters, which implies that the training stability of Transformers is an unsolved problem. The contribution of our work is to identify, for the first time, attention entropy and its collapse as a factor correlating with training divergence. This dynamic view of training makes our analysis much different from other strategies that focuses either on initialization, or normalizing the activations. On the empirical side, $\sigma$Reparam enables more robustness and simpler training settings compared to other approaches, see Table 1, 2, Figure 2,3,5 in the updated manuscript.
>
>
> > How did you get this equation-2? Is there any intuition of this equation?
>
>
> Here is the logic for coming up with Equation 2. We first observe an empirical correlation of small attention entropy and training instability (see, eg, Figure 1). We then identify in Theorem 3.1 that attention entropy is lower bounded by the query, key projection’s spectral norm (large norm leads to small entropy). We then seek ways of increasing the entropy lower bound, which amounts to decreasing the projection’s spectral norm. We then introduce Proposition 3.2 to illustrate that the spectral norm of a regular weight matrix grows with a rate linear to the square root of its input size, which means that spectral norm grows fast for large weight matrices. In the end, we re-parameterize all weight matrices with Equation 2, where the actual spectral norm becomes $\gamma$ which has a constant update rate independent of the weight matrix’s size. We also verified this theoretical connection empirically, see, eg., Figure 3.
>
>
> > The generalization of the method. The equation-2 is valid for the linear layer. What happened to the convolution layer?
>
>
> Note that a convolutional layer can be trivially expressed as a linear layer, where the kernel matrix of shape $(d_out, d_in, kernel, kernel)$ can be reshaped to $(d_out, d_in \times kernel ^ 2)$. For vision and speech experiments we have already applied $\sigma$Reparam both to transformer blocks and convolutional layers.
>
>
> > Is this valuable reparametrization for the convolution network?
>
>
> $\sigma$Reparam is applicable to ConvNets. However since our goal is to address the entropy collapse problem of Transformers which ConvNets don’t have, we did not make that a priority in our experiments.
>
>
> > The paper does not explain the relation between attention entropy and \sigma_reparameterisation.
>
>
> Please note that the entire Section 3 is dedicated to explaining this connection.
>
> Specifically, entropy collapse occurs when the rows of the attention matrix (before softmax) have a large norm. Regularizing the spectral norm (sigma) of the key and query matrices prevents this by restricting the norm of the rows from being too large. In theorem 3.1, we provide a lower bound on the attention entropy as a function of sigma. In other words, it is sufficient for sigma to be small in order to prevent the entropy from collapsing.
>
> Proposition 3.2 then explains how $\sigma$Reparam affects the update rate of the spectral norm, which makes it independent of weight matrix’s size. This slower growth of spectral norms is also verified empirically in all our experiments.
>
>
> > How does this learnable parameter contribute to the stability of the transformer?
>
>
>
> See the response to the point above, and also our general response to all reviewers on ablation with spectral norm. Note that if we do not have gamma parameter (which amounts to using spectral norm), performance of the models is very poor as it restricts model capacity.

---

> > ### Author Response · Authors · 2022-11-18
> > **Response 2/2**
> >
> > > Instead of evaluating the proposed method on multiple tasks, the author could have done an ablation analysis on different kinds of networks and various types of transformer architecture. The author should provide ablation analysis results to clarify the proposed method's work principle.
> >
> >
> > As transformer is a main architecture in many domains, we believe it is necessary to check applicability of new methods across domains to speed up research community. As transformer architectures varies across domains, we automatically cover different architecture types: encoder transformer (vision, speech) and decoder transformer (language modeling); different optimizers: adam for vision, adagrad in speech, Nesterov SGD in language modeling and lars for speech and vision, pre-layer norm (vision, speech) and post-layer norm (speech and language modeling), convolutional frontend (vision - patchifying, speech - striding) and embedding frontend (tokenization in language modeling), different positional embedding (learnable in vision, sinusoidal in language modeling, CAPE in speech).
> >
> > On ablations analysis we conducted experiments for different types of instability we observed in different domains: speech - training deep architectures out of the box and with lars instead of adam/adagrad,  robustness to hyper parameters as post-LN is very unstable to minor change of them; vision - self-supervised instability reported in prior works as discussed in the paper, simplification of training recipe for ViT as many hyper parameter engineering is necessary to have stable training.
> >
> >
> > > Why do we need such reparametrization, learning another parameter? Did you analyze other kinds of techniques to tackle the attention entropy collapse problem? This requires a substantial literature survey and motivation for the proposed method. Can you justify this? How is the attention entropy collapse solved by the proposed reparametrization technique?
> >
> >
> > We are the first who observed the entropy collapse, and it is our contribution. In the general response you can find additional experimental ablations that spectral normalization and weight normalization are not able to prevent attention entropy collapse and moreover they are not generalizable (validation performance is worse than post/pre-LN baselines and than our sigma-reparametrization). Further, we have theoretical analysis and empirical analysis which show that exactly unbounded spectral norm causes attention entropy collapse which cause training instability (see Theorem 3.1, analysis in Sec 3.2 and experiments e.g. Fig 1 and Fig 2). In short, attention entropy is lower bounded by spectral norm, which is controlled now by our reparametrization as we learn the spectral norm scalar (see Eq. 1).
> >
> >
> > > The author should compare the stability analysis results of this transformer with the existing state-of-the-art stable transformer. Also need to add a small paragraph in the related work section about stability/robustness in the transformers.
> >
> >
> > Our empirical and theoretical analysis that attention entropy collapse is one of the sources of transformer instability do not exist in prior work. The layer-norm variants  we used in the paper are the state-of-the-art transformers in every domain. In related works we discussed the works on modifications people did to remove layer-norm (e.g. ReZero, specific initialization schemes). Could Reviewer be more specific and point to the works which we missed in the paper? Note, stability and robustness we mean in the context of hyper-parameters changes (e.g. learning rate, see Figure 1).
> >
> >
> > > Can you elaborate on the number of parameters and FLOPS required for this? What about the complexity of the proposed method?
> >
> >
> > During inference $\sigma$Reparam has no additional cost over regular linear layers as Equation 2 can be fused to a single matrix. Plus, because $\sigma$Reparam enables removing Layer Norms, we can automatic efficiency gains. For example, we observe a %7 saving in both time and memory with the ViT-B model reported in Table 1. During training, as already explained in Section 3.2, the computational cost is small as the power iteration is only performed on weights rather than activations, which means that its overhead vanishes as batch size and input size increases. In practice, we do not see any changes in speed too as computation of spectral norm is done via one step of power method. We provide training speed measured for speech experiments where all combinations were tested (time is given per 1 update on A100 8GPUs with ~290ms batch per GPU): 450ms for pre-LN, post-LN, sigmaReparam, while 510ms for sigmaReparam + post-LN.
> >
> >
> > > The name spectral reparametrization in the title makes it confusing here. It seems it works on frequency domain reparametrization.
> >
> >
> > Please note that spectral (and sigma) is a standard mathematical notation for matrix norm, see https://en.wikipedia.org/wiki/Matrix_norm.
> >
> > Authors.

---

### Official Review · Reviewer_fFS8 · 2022-10-24

**Confidence:** 4
**Correctness:** 2
**Technical Novelty And Significance:** 2
**Empirical Novelty And Significance:** 2
**Recommendation:** 3

**Clarity, Quality, Novelty And Reproducibility:**

Weight norm is mentioned, which sounds very similar from the motivation to decouple the magnitude of the weights. It is claimed that its effectiveness in Transformers is limited. Was that tested? Where are the results? It would be interesting to see such a comparison.

I'm not so familiar with image classification. ImageNet-1k current SOTA seems to be 88.3% for Top1 accuracy (https://paperswithcode.com/sota/image-classification-on-imagenet?tag_filter=171). The provided 82.2% are quite far away from that?

The table 1 should contain some recent SOTA results.

Why only the train-clean-100 subset of Librispeech? This subset is way too easy, way too small, and not really used often in the literature, so not good for comparison. It should be tested on the real standard Librispeech, and potential other relevant speech corpora.

"We use SpecAugment (Park et al., 2019) only activated right at the beginning of training." - What does this mean? Later in training it is deactivated?

"Training We use Adagrad (Duchi et al., 2011) if not specified otherwise," - why Adagrad? This is quite unusual. How does it compare to SGD or Adam?

In Table 3, there should be other results from the literature for comparison. Esp, it is said that the model is used from (Likhomanenko et al., 2021a). Where is that in the table? The numbers in (Likhomanenko et al., 2021a) all look better?

I don't exactly understand the speech recognition results. So it seems that in general, the proposed method yields worse results than the baseline?

**Strength And Weaknesses:**

Strength:
- Somewhat novel reparameterization of the weights.
- Some analysis on attention entropy and its correlation to the training stability.

Weaknesses:
- Very similar to weight norm, so the novelty is limited.
- No actual comparisons to weight norm?
- Source code not released?
- Results are not good?
- Librispeech train-clean-100 is not representative as a speech recognition task.


**Summary Of The Paper:**

Setting: Transformer model, test on speech recognition and other tasks.

Measure attention entropy.
Find correlation between the minima of attention entropy and the model’s training stability.
(How is training stability measured?)
Reparametrize all linear layers with Spectral Normalization and an additional learned scalar.

Experiments with σReparam on image classification, image self supervised learning, automatic speech recognition and language modeling.


**Summary Of The Review:**

Unfortunately there are too many weaknesses which should be addressed. If those are addressed, and it can indeed yield improvements over good SOTA models, on actual relevant benchmarks, then this would be very interesting. But this requires much more work.

If the argument is only about more stable training and not about better performance, this anyway needs to be better justified and tested.

---

> ### Author Response · Authors · 2022-11-18
> **Response 1/2**
>
> Dear Reviewer fFS8,
>
> Thanks for your time and valuable feedback. Please find below our responses to your comments and suggestions:
>
>
> > How is training stability measured?
>
>
> Training stability can be measured by observing the training curves. Two representative patterns of instability is either divergence (see bottom right plot of Figure 1) or repeated collapse of training curves (see right half of Figure 3).
>
>
> > Weight norm is mentioned, which sounds very similar from the motivation to decouple the magnitude of the weights. It is claimed that its effectiveness in Transformers is limited. Was that tested? Where are the results? It would be interesting to see such a comparison.
>
>
> Please check the general response to all Reviewers where we address this question.
>
>
> > I'm not so familiar with image classification. ImageNet-1k current SOTA seems to be 88.3% for Top1 accuracy (https://paperswithcode.com/sota/image-classification-on-imagenet?tag_filter=171). The provided 82.2% are quite far away from that? The table 1 should contain some recent SOTA results.
>
>
>
> Provided reference includes results obtained with large model (ViT-H), high resolutional inputs (448x448 images vs 224x224), different training method (sophisticated self-supervised pretraining), long training epochs etc, which make them not comparable with our results.
>
> In our experiments, we focused on standard backbones (ViT-B, ViT-L) and standard training settings (eg., with optimized data augmentation and regularization schemes). All experiments across all the tasks are conducted in the control experiment settings, which we believe is a reasonable way of highlighting our main contribution while eliminating other irrelevant factors.
>
>
> > Why only the train-clean-100 subset of Librispeech? This subset is way too easy, way too small, and not really used often in the literature, so not good for comparison. It should be tested on the real standard Librispeech, and potential other relevant speech corpora.
>
>
>
> One of the active research in speech is self-supervised and semi-supervised training where 100h is one of the benchmarks (it is used as supervised and the rest of Librispeech or LibriLight is used as unsupervised data, e.g. wav2vec 2.0, slimIPL). Also we want to probe our method in the regime of less data as transformers are data hungry. Moreover, in this way we are in the regime of distributional shift as 100h contains only clean speech while validation has noisy speech. To clarify applicability to larger data we run experiments on the full LibriSpeech using two schedules on learning rate: cosine and step-wise decay with total 500k updates and adagrad optimizer. The only change we did is decreasing dropout to 0.1 per (Likhomanenko et al., 2021a). We used same parameters as for 100h setting and only searched over learning rates. We again observed post-LN instability by changing learning rate a bit, while pre-LN and $\sigma$Reparam are stable. Results are attached below and can also be found in updated version, Appendix D2 Table 7 and 8.
>
> | model  | lr schedule   | dev-clean  |  dev-other  | test-clean  | test-other  |
> |---|---|---|---|---|---|
> | https://arxiv.org/pdf/2010.11745.pdf (relpos)	| step-wise	| 2.6	| 7	| 2.7	| 6.8 |
> | post-LN (CAPE)	| cosine	| 2.6	| 7.1	| 2.9	| 7.2|
> | pre-LN  (CAPE)	| cosine	| 2.6	| 6.9	| 2.8	| 7|
> | sigma (CAPE)	|cosine	| 2.8	| 7.6	| 3	|7.7|
> |sigma + post-LN (CAPE)	|cosine|	2.7	|7.3|	2.9	|7.2|
> |post-LN (CAPE)|	step-wise|	2.6	|6.9|	2.7|	6.9|
> |pre-LN (CAPE)|	step-wise	|2.9	|7.7|	3	|7.8|
> |pre-LN (CAPE), optimized|	step-wise	|2.6	|6.8	|2.8	|6.8|
> |sigma (CAPE)|	step-wise|	2.7|	7.2|	2.9|	7.3|
> |sigma + post-LN (CAPE)|	step-wise|	2.8	|7.1	|2.9|	7|
>
>
> > "We use SpecAugment (Park et al., 2019) only activated right at the beginning of training." - What does this mean? Later in training it is deactivated?
>
>
>
> We apologize for the confusion caused. SpecAugment is activated right at the beginning, meaning it is used at every training step. We will update the text.
>
>
> > "Training We use Adagrad (Duchi et al., 2011) if not specified otherwise," - why Adagrad? This is quite unusual. How does it compare to SGD or Adam?
>
>
> SGD doesn’t work with post-LN in any scenario, while for pre-LN and sigma-reparametrization it has significantly worse train loss and generalization. We experimented with Adam but results are worse than with Adagrad (in terms of validation WER). Also prior works from which we use current baselines (state-of-the-art for CTC) reported Adagrad as optimizer (see e.g. Likhomanenko et al., 2021a).

---

> > ### Author Response · Authors · 2022-11-18
> > **Response 2/2**
> >
> > > In Table 3, there should be other results from the literature for comparison. Esp, it is said that the model is used from (Likhomanenko et al., 2021a). Where is that in the table? The numbers in (Likhomanenko et al., 2021a) all look better?
> > I don't exactly understand the speech recognition results. So it seems that in general, the proposed method yields worse results than the baseline?
> >
> >
> > Thank you for the suggestion, we will add the baseline from Likhomanenko et al., 2021a into the Table. To clarify (which is in the main text) compared to Likhomanenko et al., 2021a we do not use learnable relative positional embedding as it is expensive: 2-3x slow down per update and needs more memory. With respect to our stability and robustness analysis we switched to recently proposed by Likhomanenko et al., 2021b CAPE embedding which is an absolute embedding, cheap and can model relative positional embedding (was tested on Tedlium and WSJ datasets). $\sigma$Reparam reaches similar performance as corresponding vanilla transformer with post-LN which also uses the same positional embedding.
> >
> >
> > > If those are addressed, and it can indeed yield improvements over good SOTA models, on actual relevant benchmarks, then this would be very interesting.
> >
> >
> > All our experiments in Vision (supervised, self supervised), Speech and Language are based on highly optimized baselines that are recent and widely cited. We believe that these are relevant baselines in their respective settings, and they are standard benchmarks for measuring progress in each respective field.
> >
> >
> > > If the argument is only about more stable training and not about better performance, this anyway needs to be better justified and tested.
> >
> >
> > On the empirical side, we are able to match or outperform standard baselines. Second, in all our settings, we are able to train the models without relying on Layer Norm, which is a core component for stable training. This gives us better training robustness as already shown in Figure 3, 5, as well as Section 4.1.
> >
> >
> > > Source code not released?
> >
> >
> > Note that $\sigma$Reparam is a plug in module to existing Transformer baselines and we already provided its pseudo code in Algorithm 1. We have also attached a Pytorch implementation of the Linear layers which can be used to reproduce our results.
> >
> > Authors.

---

### Official Review · Reviewer_janJ · 2022-10-25

**Confidence:** 2
**Correctness:** 4
**Technical Novelty And Significance:** 3
**Empirical Novelty And Significance:** 3
**Recommendation:** 6

**Clarity, Quality, Novelty And Reproducibility:**

Clarity

The paper is quite clear

Quality

The paper is high quality in terms of the experiments. The argument motivating of $\sigma$-reparameterisation from the standpoint of fixing the problem of entropy collapse could be improved. However I wouldn't classify this as the main contribution.

Novelty

As far as I am aware, the method is novel. However, I am not an expert on transformer regularization, and this approach could well be discussed in an appendix of previous work or similar.

Reproducibility

The code does not seem to be available, but the method is so simple it could be implemented and reproduced quite quickly (modulo the expense of training a large transformer model nowadays)

**Strength And Weaknesses:**

Strengths
+ The empirical results are strong, showing increased robustness to changing hyperparameters and similar or better final performance. This is a good result for democratisation of ML, since it reduces the amount of time and money that must be spent on fiddly hyperparameter tuning. Figure 4 is particularly compelling.
+ The method is extremely simple to implement, yet has (to my knowledge) not been proposed before.
+ The experimental results are fairly comprehensive.

Weaknesses
+ While I believe the final experimental results stand on their own, on a scientific level I am not convinced by the argument relating the failure to train to the collapse of the attention-entropy. In the case in figure 1, the two phenomena are definitely correlated, but this doesn't appear to continue in figure 2, where the baseline attention entropy 'collapses' but still gives good downstream performance. Similarly, none of the successful baseline models in figure 3 seem to have this 'dip' phenomenon where the entropy decreases then increases as the model begins to learn properly. Similarly, the connection between the singular values and attention entropy doesn't seem to hold practically: look at figure 3 where we have the green solid and dotted line have similar singular values but completely different attention entropy. It seems plausible that the collapse of the attention entropy and the failure of downstream performance could both be symptoms of some other problem, which the $\sigma$-reparameterisation avoids. I realise the authors are careful to not say that entropy collapse causes training instability, but I believe it could be possible to get some direct evidence for this claim, which would strengthen the scientific contribution of the paper.

  + Nitpicks:
    + Could you try a baseline of dividing the initalization by a factor of the spectral norm of a random matrix?
    + In the theory you use $\|W_KW_Q^\top\|_2$, but in the implementation it seems that you apply the reparameterisation on all of the weight matrices. Did you try only reparameterising the $W_KW_Q^\top$ matrix directly?
    + How does the overall memory usage compare when using LARS and fp32 on attention compared to adam and full mixed precision?


**Summary Of The Paper:**

The paper proposes the $\sigma$-reparameterised transformer. The authors note a phenomenon in during training of transformers where the 'attention entropy' (essentially, how sharply peaked the attention scores are) decreases and then increases again. In the cases where the transformer fails to learn, this attention entropy does not recover. In order to combat this, the $\sigma$-reparameterisation is introduced, essentially a version of spectral normalization where the matrices are rescaled by a learned parameter (instead of having unit norm). This provides impressive gains in stability of training and empirical performance on a variety of tasks.

**Summary Of The Review:**

The paper is a good empirical contribution to the state of the art of training transformer-based models, giving a new form of parameterisation of the weight matrices that dramatically improves the robustness to hyperparameter choices. This will lead to improved reliability of training transformers. The scientific argument analysing the attention collapse is a bit weak, and could be improved. But since that is not the focus of the paper, it shouldn't be required for an accept.

Update after rebuttal:
After reading the authors' rebuttal, I believe they have addressed my queries and some of the insightful remarks from the other reviewers.

Update after end of rebuttal period:
Given the lack of response by the reviewers to the rebuttal to the experimental queries from the other reviewers, I'm unsure to what extent the issues raised by the other reviewers have been satisfied. Since I'm not completely familiar with the state of the art in stable transformer architectures, I've reduced my confidence.

---

> ### Author Response · Authors · 2022-11-18
> **Response**
>
> Dear Reviewer janJ,
>
> Thanks for your time and recognition of our contribution! Please find below our comments.
>
>
> > ...but this doesn't appear to continue in figure 2, where the baseline attention entropy 'collapses' but still gives good downstream performance
>
>
> We apologize for the confusion. The baseline shown in Figure 2 is actually a stable training run that does not suffer from entropy collapse, obtained with extensive hyper parameter tuning (learning rate, warmup epochs, batch size, optimizer momentum, weight decay). This can be understood by looking at the absolute values of the attention entropy, which is maintained around 2.5. Because attention entropy is bounded between (0, log(T)] (where log(T) = 5.28 for the ViT experiments), we refer to entropy collapse as the phenomenon where entropy drops to near zero, as shown in Figure 1. We have updated the text in the paper to make this definition more explicit.
>
>
> > Similarly, none of the successful baseline models in figure 3 seem to have this 'dip' phenomenon where the entropy decreases then increases as the model begins to learn properly.
>
>
> While the exact shape of the attention curves in Figure 3 differ from those in Figure 1, we see that the dips of the entropy curves correlate very well with the accuracy curve for the baseline models (the red and green curves). We thus believe that these curves are consistent with Figure 1.
>
>
> > Similarly, the connection between the singular values and attention entropy doesn't seem to hold practically: look at figure 3 where we have the green solid and dotted line have similar singular values but completely different attention entropy.
>
>
> We’d like to point out that the $\sigma$-norm plots were originally in the log scale, where the sigma values of the green and solid lines appear similar. In the updated manuscript, we have re-generated the same plots in linear scale to emphasize where they diverge. What we see is that the sigma values the green solid and dashed curves diverge in a manner that tracks the attention entropy curves very tightly. Specifically, at around epoch 20, we see that the dashed green line attention entropy drops to a low value, coinciding with an increase in its spectral norm. This is a clean example of the attention entropy collapse phenomenon, and is a behavior not shared by the solid green run.
>
>
> > It seems plausible that the collapse of the attention entropy and the failure of downstream performance could both be symptoms of some other problem, which the σ-reparameterisation avoids
>
>
> We agree that there might be other reasons causing training instability, and it remains an open problem as of how exactly entropy collapse contributes to training instability. However, we do believe that there is enough evidence that the attention entropy has strong correlation with training stability, and that bounding it (via $\sigma$Reparam ) is beneficial for training stability. We will incorporate the feedback and include more explicit discussions.
>
>
> > Could you try a baseline of dividing the initalization by a factor of the spectral norm of a random matrix?
>
>
> Yes. This is one of the things we attempted after first observing entropy collapse - unfortunately it does not prevent the collapse.
>
>
> > In the theory you use |WKWQ⊤|2, but in the implementation it seems that you apply the reparameterisation on all of the weight matrices. Did you try only reparameterising the WKWQ⊤ matrix directly?
>
>
> Yes, we did try this and it does not provide the full training stability without the presence of Layer Norm. The reason is that the attention entropy also decreases with large activation norms, which can become unbounded without the use of Layer Norm. Applying $\sigma$Reparam everywhere is a convenient technique to also slow down the growth of activation norms, which also simplifies the implementation.
>
>
> > How does the overall memory usage compare when using LARS and fp32 on attention compared to adam and full mixed precision?
>
>
> For speech experiments we trained models on A100 with tensor cores which are somewhat bfloat16 and did not observe any precision issues. Language model experiments are run in fp32. For vision experiments we faced some precision issues for mixed precision training which we believe are related to ViT specifics. fp32 does introduce more memory cost with LARS, compared to ADAM with AMP, however we believe that the precision issue is an engineering aspect that can be optimized independently.
>
> Compared to Adam, LARS is cheap as we do not store gradients momentums, we only normalize gradients of every matrix by its norm, which is an inplace operation. You can think about LARS as standard SGD in terms of used memory.
>
>
> > As far as I am aware, the method is novel. However, I am not an expert on transformer regularization, and this approach could well be discussed in an appendix of previous work or similar.
>
>
> We will update related works section.
>
> Authors.

---

### Official Review · Reviewer_3X2m · 2022-10-29

**Confidence:** 4
**Correctness:** 2
**Technical Novelty And Significance:** 2
**Empirical Novelty And Significance:** 2
**Recommendation:** 3

**Clarity, Quality, Novelty And Reproducibility:**

(-) The paper is not clear. It introduces a spectral normalization, but the explanation of how it differs from past spectral normalization is not convincingly given.

(-) Many normalization methods have been proposed, and it is not clear how this differs from the past such as Eq(2). The paper discusses briefly the differences, but the constant gamma in the normalization does not seem to be a major difference.

(+) The paper should be reproducible but it is not certain.

**Strength And Weaknesses:**

(-) The readability of the paper needs much to be desired. It seems that instability is measured by the entropy of the attention head. Decreasing the learning rate or increasing warmup epoch leads to more stability. Without providing any explanation, the paper mentions that the training loss decreases with increase in learning rate. The paper should state whether decrease in training loss comes from stability. What is the consequence of entropy collapse? Explain how entropy collapse relates with the stability and accuracy? Is  stability and accuracy related? It is not clear how the lower bound of the attention entropy is related to stability.

(-) The paper is proposing a re-parameterization of the weights of the linear layer using a spectral norm. It is not clear how this is related to regularization and performance increase? Many normalization methods have been proposed in the past, and it is not clear how the proposed is different previous normalization methods- a table should compare the performance of previous methods.

**Summary Of The Paper:**

This paper studies the instability of Transformers by studying the progression of attention entropy: first a decrease followed by a quick increase then a long stable phase. A strong correlation exists between the minima of attention entropy and training stability. In the study, decreasing learning rate and increasing the warmup epoch leads to lower training loss. In addition, the following observations were made: Increasing the learning rate can lead to attention entropy collapse.To lower attention entropy,  reparametrizement of all linear layers with spectral normalization is considered.  The proposed algorithm is evaluated on 4 tasks: image classification, image self-supervised learning, ASR, and language modeling.

**Summary Of The Review:**

See strength and weakness.

---

> ### Author Response · Authors · 2022-11-18
> **Response 1/2**
>
> Dear Reviewer 3X2m,
>
> Thanks for your time and valuable feedback.
>
>
> ### Relation between training stability, training loss and generalization.
>
>
> We apologize for the confusions. To clarify, we first show an ablation study of a baseline ViT model trained on ImageNet (corresponding to Figure 1 in the paper)
>
> |   | baseline  | lr x 2  | lr x 0.4  | lr x 0.2  | warmup x 0.5  |  warmup x 2 |  warmup x 4 |
> |---|---|---|---|---|---|---|---|
> | train loss  |   2.59 | diverge  | 2.59  |  2.53  |  diverge  |   2.48	  |   2.47 |
> | test accuracy %  |  81.8  |  diverge | 78.9  | 76.0  | diverge  |  79.1  |  78.9 |
>
>
> We see that both lowering the learning rate and increasing the number of warmup epochs lead to slightly smaller training losses and smoother training curves (see Figure 1). However, both smaller learning rate and longer warmups lead to worse test accuracy. Further increase learning rate or decreasing warmups from the optimal setting (the baseline) leads to training divergence.  With this in mind, we can answer the specific questions:
>
>
> > Without providing any explanation, the paper mentions that the training loss decreases with increase in learning rate. The paper should state whether decrease in training loss comes from stability.
>
>
> First, in the particular example above, increasing the LR leads to divergence, while decreasing LR leads to slightly reduced training loss, we are assuming that the Reviewer made a typo here. As shown in Figure 1, smaller LR leads to significantly larger attention entropy through training, which also results in smoother training loss curves. That being said, the LR itself also has a non-trivial impact on convergence even in the stable regime.
>
>
> > What is the consequence of entropy collapse?
>
>
> Entropy collapse denotes the phase where the attention entropy of all layers drop to near zero, as shown in the bottom right plot of Figure 1. Also note that the attention entropy by definition is bounded in (0, log(T)]. When entropy collapse happens, training diverges and loss stops decreasing. Empirically, we have seen that near zero attention entropy (e.g. entropy collapse) almost always leads to model divergence (also see Figure 3 for more evidence).
>
>
> > Explain how entropy collapse relates with the stability and accuracy? Is stability and accuracy related?
>
>
> We have explained above that entropy collapse is correlated with training stability. Training stability as we define it in the paper is a training property, and does not directly affect generalization as far as we can tell. However, stability allows one to train with a wider range of hyperparameters (e.g. large LR, short warmups, different initializations, weight decay, see Table 1, 2 Figure 5), potentially achieving better performance). In particular, we have found that large learning rates are beneficial for generalization, which however often leads to instability, see the ablation table above. Also see existing works on the relation between large LR and generalization [1, 2].
>
>
> >  It is not clear how the lower bound of the attention entropy is related to stability
>
>
> We do not prove a causal relationship between entropy collapse and unstable behavior. We do see however a clear correlation between the two. Our hypothesis is that by preventing the entropy from collapsing, unstable training can be avoided, which is what we empirically verify in the paper. The lower bound of attention entropy dictates the smallest entropy achievable by an attention layer. Increasing the lower bound thus would prevent attention collapse from happening. See, e.g., Figure 3 for an illustration.
>
>
> > The paper is proposing a re-parameterization of the weights of the linear layer using a spectral norm. It is not clear how this is related to regularization and performance increase?
>
>
> First of all, $\sigma$Reparam does not constrain the model space due to its re-parameterization nature. We thus do not expect it to provide an explicit regularization (compared to e.g. Spectral Norm [3] which constraints the Lipschitz constant of a layer). As stated above, the benefits of $\sigma$Reparam include 1) increasing model’s robustness wrt hyper parameters and 2) enabling large learning rates which infeasible with the standard Transformers.

---

> > ### Author Response · Authors · 2022-11-18
> > **Response 2/2**
> >
> > > Many normalization methods have been proposed in the past, and it is not clear how the proposed is different previous normalization methods- a table should compare the performance of previous methods.
> >
> >
> > Thanks for the suggestion. We have included comparisons with both Weight Norm [4] and Spectral Norm [3] in the updated Table 1, 3, 5. We see that $\sigma$Reparam outperforms both. In particular, Weight Norm does not effectively constrain the attention entropy, as discussed in Sec 3.1 of the paper. Spectral Norm on the other hand severely constrains the model’s capacity which also results in large performance gaps. See the general response for more details.
> >
> > Also note that a large part of the contribution of this paper is identifying the entropy collapse problem as symptom of training instability, establishing a theoretical connection to the model’s training dynamics and promising a simple solution $\sigma$Reparam. $\sigma$Reparam differs from both Weight Norm and Spectral Norm in subtle but important aspects, which is verified both theoretically and empirically.
> >
> > References:
> >
> > [1] The large learning rate phase of deep learning: the catapult mechanism, Lewkowycz et al, 2020
> >
> > [2] Understanding Gradient Descent on the Edge of Stability in Deep Learning, Arora et al, ICML 2022
> >
> > [3] Spectral Normalization for Generative Adversarial Networks, Miyato et al, ICLR 2018
> >
> > [4] Weight Normalization: A Simple Reparameterization to Accelerate Training of Deep Neural Networks, Salimans & Kingma, 2016

---

### Decision · Program_Chairs · 2023-01-20

**Decision:**

Reject

**Justification For Why Not Higher Score:**

Three out of four ratings are reject, readability should be improved, unclear whether training instabilities really caused by attention entropy.

**Justification For Why Not Lower Score:**

N/A

**Metareview: Summary, Strengths And Weaknesses:**

Reviewer 3X2m. Rating: 4, Conf: 4.
Reviewer janJ. Rating: 6, Conf: 2.
Reviewer fFS8. Rating: 3, Conf: 4.
Reviewer qd4k. Rating: 3, Conf: 4.

The paper shows during transformer training, the attention entropy typically decreases significantly, before increasing significantly again. They hypothesize that the lowest point of entropy correlates with training stability (and back this with an experiment that shows a correlation between dips in entropy and instability). They then propose to lower bound the entropy through a new method, sigma-reparam. Experiments show that the method results in impoved training stability and robustness.

Reviewers note that the empirical results are fairly comprehensive and strong, and that the method is simple to implement.
Reviewers also note that readability should be improved, and that more comparisons should be done with existing regularizers. As a response, the authors include comparisons to Weight Norm and Spectral Norm. Mot importantly, the authors failed to provide convinving proof that attention entropy causes optimization instabilities.
In general, the authors did a good job responding to the reviewers' critiques. Unfortunately the reviewers declined to further engage, and did change their scores, except for Reviewer janJ who reduced his score from 8 to 6 and confidence from 3 to 2 after the rebuttal period.

On Nov 17, the authors wrote that they found the reviews by reviewers fFS8 and qd4k to be low quality and unfair. However, even when ignoring these two reviews, the score would IMO still be too low for acceptance.